# *ReSearch*: Learning to *Re*ason with *Search* for LLMs via Reinforcement Learning

**Mingyang Chen**[1], **Linzhuang Sun**[2], **Tianpeng Li**[1], **Haoze Sun**[1], **Yijie Zhou**[1],
**Chenzheng Zhu**[1], **Haofen Wang**[3], **Jeff Z. Pan**[4], **Wen Zhang**[5], **Huajun Chen**[5],
**Fan Yang**[1]*, **Zenan Zhou**[1], **Weipeng Chen**[1]

[1]Baichuan Inc. [2]University of Chinese Academy of Sciences
[3]Tongji University [4]The University of Edinburgh [5]Zhejiang University
{chenmingyang, yangfan}@baichuan-inc.com
https://github.com/Agent-RL/ReSearch

## Abstract

Large Language Models (LLMs) have shown remarkable capabilities in reasoning, exemplified by the success of OpenAI-o1 and DeepSeek-R1. However, integrating reasoning with external search processes remains challenging, especially for complex multi-hop questions requiring multiple retrieval steps. We propose *ReSearch*, a novel framework that trains LLMs to *Re*ason with *Search* via reinforcement learning without using any supervised data on reasoning steps. Our approach treats search operations as integral components of the reasoning chain, where when and how to perform searches is guided by text-based thinking, and search results subsequently influence further reasoning. We train *ReSearch* on Qwen2.5-7B(-Instruct) and Qwen2.5-32B(-Instruct) models and conduct extensive experiments. Despite being trained on only one dataset, our models demonstrate strong generalizability across various benchmarks. Analysis reveals that *ReSearch* naturally elicits advanced reasoning capabilities such as reflection and self-correction during the reinforcement learning process.

## 1 Introduction

In recent years, Large Language Models (LLMs) have demonstrated remarkable performance across a wide array of tasks [1, 5, 13, 39]. Beyond leveraging internal knowledge acquired during pretraining, LLMs exhibit the capability to utilize external tools, particularly search engines, to retrieve factual and time-sensitive information, thereby mitigating instances of hallucination [4, 14, 21, 25]. This capability, often referred to as Retrieval-Augmented Generation (RAG), has been the subject of extensive investigation in recent literature [2, 6, 38, 42]. Despite the effectiveness of RAG, designing robust multi-step RAG strategies applicable to complex real-world problems remains a significant challenge. This is particularly crucial, as many real-world issues are inherently complex and necessitate several steps of reasoning [23, 33, 35].

The past year has witnessed considerable advancements in LLMs' reasoning abilities, particularly through chain-like reasoning before producing final outputs [37, 41]. This progress is exemplified by the success of OpenAI-o1 [17], and DeepSeek-R1 [5]. These developments emphasize the importance of test-time scaling in reasoning, enabling LLMs to decompose intricate problems into manageable intermediate steps [16, 28]. This reasoning capacity is also vital for the efficacy of RAG, especially when addressing complex questions that require multiple retrieval steps. Nonetheless, training LLMs

---

*Corresponding author

39th Conference on Neural Information Processing Systems (NeurIPS 2025).

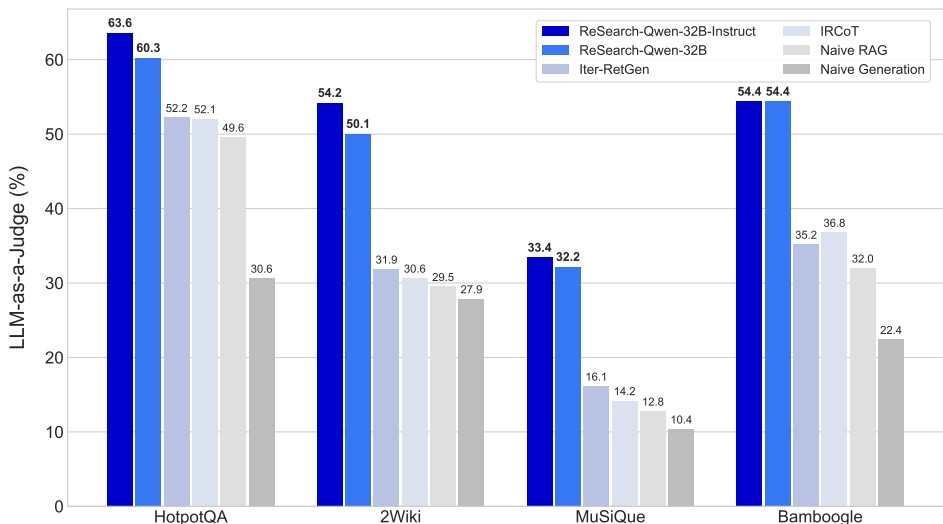

Figure 1: Comparative performance of *ReSearch* and baseline methods on benchmark datasets. All baselines are built upon Qwen2.5-32B-Instruct. See Section 3 for details.

to conduct interactive reasoning alongside information retrieval continues to present an open challenge for the research community. Most existing approaches to multi-step RAG rely on manually designed prompts or heuristics, which are not only labor-intensive but also lack scalability for more intricate problems [19, 23, 33]. Additionally, labeling reasoning steps in a multi-step RAG framework is often impractical due to the associated costs and time constraints.

Reinforcement learning (RL) has emerged as a promising avenue for enhancing reasoning capabilities without the need for supervised data regarding reasoning steps [5, 24]. This approach holds potential for training LLMs to exhibit reasoning skills solely based on simple reward signals derived from final outcomes. Recent advancements in RL-based training for LLMs have demonstrated significant improvements in complex reasoning tasks, where models learn to decompose problems into manageable steps through trial and error rather than explicit instruction. Models such as DeepSeek-R1 have shown that rule-based reward functions can effectively guide LLMs to develop sophisticated reasoning patterns autonomously. Despite these successes, current approaches primarily focus on enhancing internal reasoning capabilities, with limited exploration of how to effectively combine this reasoning process with external knowledge retrieval.

In this paper, we propose a novel framework for training LLMs to *Re*ason with *Search* via reinforcement learning, which we term **ReSearch**. The reasoning chain in this framework is not only composed of text-based thinking (i.e., enclosed by `<think> </think>`) as DeepSeek-R1, but also search query (i.e., enclosed by `<search> </search>`) and retrieval results (i.e., enclosed by `<result> </result>`). We treat the search operation as part of the chain-like reasoning process, and the search operation will interact with text-based thinking. Specifically, when and how to perform search will be steered by previous text-based thinking and the search results will infuence subsequent text-based thinking. In the framework, we don't provide any supervised data on reasoning steps for LLMs to imitate, instead, we leverage reinforcement learning (i.e., GRPO) to incentivize LLMs to perform reasoning with search.

We train *ReSearch* from scratch on Qwen2.5-7B(-Instruct) and Qwen2.5-32B(-Instruct), and conduct extensive experiments on multi-hop question answering benchmarks that need multi-step reasoning and multiple information retrieval. Our trained models show significant absolute improvements range from 8.9% to 22.4% over the baselines, as shown in Figure 1. Furthermore, our training is only conducted on one specific training set, and trained models are evaluated on multiple benchmarks, showing the generalizability of our framework. Our contributions are as follows:

- By emphasizing the interaction between reasoning and search, we propose a novel framework *ReSearch* that using reinforcement learning to train LLMs to reason with search from scratch, without any supervised data on reasoning steps.

- We train *ReSearch* on different scales of models, and conduct extensive experiments on multi-hop question answering benchmarks, showing the effectiveness of this framework. The trained models show significant generalizability and potential for more realistic scenarios.

- By analyzing the training process, we demonstrate that *ReSearch* can effectively elicit reasoning capabilities with search progressively itself, and that reasoning abilities such as reflection and self-correction can be incentivized without relying on any pre-defined heuristics.

## 2   Method

Drawing inspiration from the success of OpenAI-o1 and DeepSeek-R1 in learning to reason, we incorporate search operation into the reasoning process and train LLMs from scratch using reinforcement learning (i.e., GRPO) without any labeled data on reasoning chains, making LLMs learn to *Re*ason with *Search* (*ReSearch*). In this section, we first show the details of training *ReSearch*, dive into the details of the GRPO and how to conduct rollout with search during reinforcement learning (§2.1). Then, we demonstrate the prompt template design directing the LLMs to generate the defined format of rollout (§2.2), and finally, we introduce the reward modeling for guiding the optimization of reinforcement learning (§2.3).

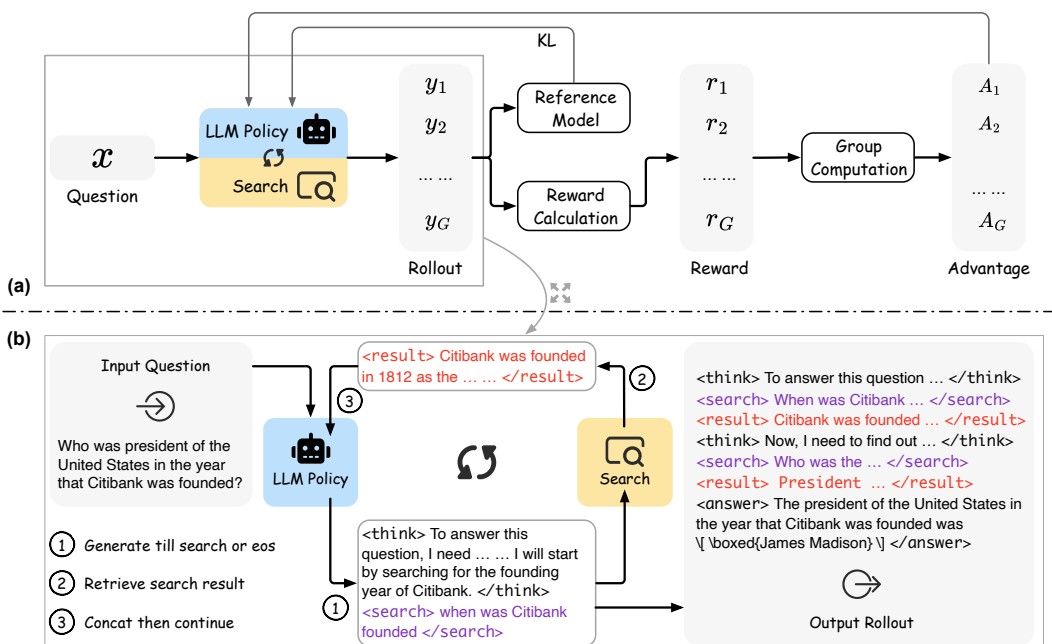

Figure 2: The training overview of *ReSearch*. (a) The GRPO pipeline. (b) The details of the rollout generation process.

### 2.1   Reinforcement Learning

When handling complex multi-step tasks needing retrieval (i.e., multi-step RAG), reasoning is crucial for steering multiple retrieval (i.e., search) operations, mainly on when and how to perform search. It's challenging to collect labeled reasoning data with search for supervised fine-tuning LLMs to imitate how to reason with search. Fortunately, reinforcement learning has shown impressive performance in training LLMs to conduct reasoning, which can elicit reasoning capabilities from LLMs without any supervised data. In general, the main idea behind reinforcement learning here is to sample multiple reasoning-with-search chains (i.e., rollouts) and optimize the policy (i.e., LLMs) to maximize the probability of generating rollouts with higher rewards, as described in Figure 2.

**Group Relative Policy Optimization**   Specifically, in this work, we use Group Relative Policy Optimization (GRPO) as the learning algorithm, which estimate the baseline from a group of rollouts

instead of training a separate critic model in Proximal Policy Optimization (PPO). Given an existing policy $\pi_{\theta_{\text{old}}}$ and an reference policy $\pi_{\theta_{\text{ref}}}$, base on $G$ rollouts $\tau = \{y_i\}_{i=1}^G \sim \pi_{\theta_{\text{old}}}(\cdot|x)$ for each input $x \sim \mathcal{D}$, the objective of GRPO is to optimize the policy $\pi_\theta$ by maximizing the following objective:

$$\mathcal{J}(\theta) = \mathbb{E}_{x \sim \mathcal{D}, \{y_i\}_{i=1}^G \sim \pi_{\theta_{\text{old}}}(\cdot|x)}$$
$$\frac{1}{G} \sum_{i=1}^G \left[ \min \left( \frac{\pi_\theta(y_i|x)}{\pi_{\theta_{\text{old}}}(y_i|x)} A_i, \text{clip}\left( \frac{\pi_\theta(y_i|x)}{\pi_{\theta_{\text{old}}}(y_i|x)}, 1-\epsilon, 1+\epsilon \right) A_i \right) - \beta \mathbb{D}_{\text{KL}}\left( \pi_\theta || \pi_{\theta_{\text{ref}}} \right) \right], \quad (1)$$

where $A_i = \left( r_i - \text{mean}(\{r_j\}_{j=1}^G) \right) / \text{std}(\{r_j\}_{j=1}^G)$ is the normalized advantage of the $i$-th rollout in current group, $\epsilon$ is the clipping ratio, and $\beta$ is the KL loss coefficient. Moreover, a KL divergence penalty is added to the objective to prevent the policy from deviating too much from the original reference policy LLMs. The illustration of GRPO is shown in Figure 2(a).

**Rollout with Search**  Compared with conventional rollout that only contains text-based thinking as reasoning, the rollout in *ReSearch* also contains search queries and retrieval results. We use `<search>` and `</search>` to enclose the search queries and `<result>` and `</result>` to enclose the retrieval results, and such instruction is described in the prompt templates, which will be introduced later in §2.2. The rollout process is an iterative process between text-based thinking, search queries, and retrieval results as described in Figure 2(b). Specifically, when the generation process encounters `</search>` tag, the query between the last `<search>` and current `</search>` tags will be used as the search query to retrieve relevant factual information, and the retrieval results will be enclosed by `<result>` and `</result>` tags. Then, existing rollout concated with the retrieval results will be used as the next input to generate following response iteratively, until the generation encounters end-of-sentence (eos) tag (i.e., `<endoftext>` or `<im_end>` in Qwen-2.5 Models).

**Retrieval Result Masking**  In original GRPO, the loss is calculated by all the generated tokens in the whole rollout. However, in *ReSearch*, the rollout contains retrieval results, which are not generated by the training policy, but retrieved by the search environment. Therefore, we mask the retrieval results in the loss calculation to avoid the training policy from being biased towards the retrieval results. That is, during the computation of Equation 1, we only consider the tokens in the text-based thinking and the search queries, and ignore the tokens in the retrieval results.

## 2.2  Training Template

Since we orchestrate the rollout process by identifying our defined special tags (e.g., stopping at `</search>` and transferring control to the search environment), it is crucial for policy LLMs to generate output in the defined format. To guide the LLMs in understanding this rollout format—specifically, the tags indicating when the search operation is invoked—we created two prompt templates: one for the base (i.e., pre-trained) model and another for the instruction-tuned model. As shown in Table 1, inspired by DeepSeek-R1, these templates are designed to be simple and concise, ensuring that the model can act as a natural progression during the reinforcement learning process. Specifically, for the *base model*, this template, filled with a specific user question, will be used as direct input to the LLMs. For the *instruction-tuned model*, its prompt template serves as the system prompt, utilized in conjunction with the corresponding chat template of the instruction-tuned LLM.

## 2.3  Reward Modeling

During reinforcement learning of *ReSearch*, there is no supervised reasoning data, and we only use a simple reward on rollouts to guide the optimization of LLMs. Experimentally, only rule-based reward function is enough to successfully elicit capabilities of reasoning with search for LLMs. Our reward function considers following two parts: answer reward and format reward.

- **Answer Reward**: We calculate the correctness of the final answer in `\boxed{}` and the ground truth answer via F1 score.

- **Format Reward**: We check whether the rollout correctly follows our defined format as described in the prompt templates, mainly checking the correctness of tags and existence of `\boxed{}` in the answer.

Table 1: Prompt templates for training from base model and instruction-tuned model. For the base model, prompt will be replaced with the actual question. For the instruction-tuned model, this template is used as the system prompt.

---

**Prompt Template For Base Model**

---

A conversation between User and Assistant. The user asks a question, and the assistant solves it. The assistant first thinks about the reasoning process in the mind and then provides the user with the answer. During thinking, the assistant can invoke the wikipedia search tool to search for fact information about specific topics if needed. The reasoning process and answer are enclosed within `<think> </think>` and `<answer> </answer>` tags respectively, and the search query and result are enclosed within `<search> </search>` and `<result> </result>` tags respectively. For example, `<think>` This is the reasoning process. `</think>` `<search>` search query here `</search>` `<result>` search result here `</result>` `<think>` This is the reasoning process. `</think>` `<answer>` The final answer is \boxed{answer here} `</answer>`. In the last part of the answer, the final exact answer is enclosed within \boxed{} with latex format. User: prompt. Assistant:

---

**System Prompt Template For Instruction-Tuned Model**

---

You are a helpful assistant that can solve the given question step by step with the help of the wikipedia search tool. Given a question, you need to first think about the reasoning process in the mind and then provide the answer. During thinking, you can invoke the wikipedia search tool to search for fact information about specific topics if needed. The reasoning process and answer are enclosed within `<think> </think>` and `<answer> </answer>` tags respectively, and the search query and result are enclosed within `<search> </search>` and `<result> </result>` tags respectively. For example, `<think>` This is the reasoning process. `</think>` `<search>` search query here `</search>` `<result>` search result here `</result>` `<think>` This is the reasoning process. `</think>` `<answer>` The final answer is \boxed{answer here} `</answer>`. In the last part of the answer, the final exact answer is enclosed within \boxed{} with latex format.

---

Specifically, for the final reward of a rollout:

$$r = \begin{cases} \text{f1}(a_{\text{pred}}, a_{\text{gt}}), & \text{if f1 score is not 0} \\ 0.1, & \text{if f1 score is 0 and format is correct} \\ 0, & \text{if f1 score is 0 and format is incorrect} \end{cases} \quad (2)$$

where $a_{\text{pred}}$ is the final answer in \boxed{} and $a_{\text{gt}}$ is the ground truth answer, and $\text{f1}(a_{\text{pred}}, a_{\text{gt}})$ is the F1 score between $a_{\text{pred}}$ and $a_{\text{gt}}$.

## 3 Experiments

### 3.1 Experiment Setup

To evaluate the effectiveness of *ReSearch*, we conduct extensive experiments mainly on multi-hop question answering benchmarks that need multi-step reasoning and multiple information retrieval. Our *ReSearch* is trained from Qwen2.5-7B, Qwen2.5-7B-Instruct, Qwen2.5-32B and Qwen2.5-32B-Instruct [39]. During training, we only use the data from training set of MuSiQue [32], since it has various types of multi-hop questions and constructed via fine-grained quality control.

**Benchmarks** We use four standard benchmarks on multi-hop question answering tasks, including HotpotQA [40], 2WikiMultiHopQA [7], MuSiQue [32], and Bamboogle [19]. Specifically, HotpotQA, 2WikiMultiHopQA, and MuSiQue are constructed among wikipedia or wikidata [34], via different multi-hop mining strategies with crowd-sourcing, while Bamboogle is manually constructed dataset with 2-hop questions, where all questions are sufficiently difficult to be unanswerable by a popular internet search engine. Our evaluation is conducted on the full dev set of HotpotQA, 2WikiMultiHopQA, and MuSiQue, and the test set of Bamboogle, including 7405, 12576, 2417, 125 samples respectively. Note that we discard the context documents from the original datasets for

HotpotQA, 2WikiMultiHopQA, and MuSiQue, and only use the question and answer pairs for evaluation. We use an open-ended retrieval environment based on wikipedia to retrieve the background knowledge for all the datasets, which we introduce later.

**Baselines** We first compare *ReSearch* with two naive baselines: (1) *No RAG*: Use corresponding instruction-tuned model to generate answer directly without any RAG, and (2) *Naive RAG*: A naive retrieval-based setting that concatenate the retrieval results with question and then generate answer directly. Furthermore, we also consider two approaches focusing on improving multi-step RAG: (3) *Iter-RetGen* [23]: A method synergizes retrieval and generation in an iterative manner, and (4) *IRCoT* [33]: An iterleaving method, which use retrieval and the chain-of-thought (CoT) guide each other. Since these methods are prompt-based, we use instruction-tuned models in same size as our *ReSearch* to implement them for fair comparison.

**Evaluation Metrics** For evaluate the correctness of the final answer, we first use Exact Match (*EM*) where the prediction is correct if it matches the ground truth answer exactly. However, such exact match is too strict for our setting, since the retrieval environment is open-ended and the result is described by natural language. Therefore, we also consider LLM-as-a-judge (*LJ*) for automatic evaluation, where we use gpt-4o-mini with our defined judge prompt to score the correctness of the final answer. Such judge prompt is shown in Appendix A.

**Implementation Details** We conduct our training and evaluation on Qwen2.5-7B, Qwen2.5-7B-Instruct, Qwen2.5-32B and Qwen2.5-32B-Instruct. The reinforcement learning framework is built on verl [26]. We only use the training set (19938 samples) of MuSiQue for training, and the number of training epochs is 2. The retrieval environment is based on FlashRAG [10], a standard toolkit for RAG research. We use E5-base-v2 [36] as the retriever and Wikipedia data from Dec. 2018 as the knowledge base [11]. All the corpus indexing and embedding has been preprocessed by FlashRAG. During the rollout in training and evaluation, we retrieve top-5 results for each query. For baseline methods, we use the implementation from FlashRAG. For details about model training, please refer to Appendix B.

## 3.2 Main Results

The main results of baselines and *ReSearch* are demonstrated in Table 2, and we show the methods based on LLMs with different sizes respectively. From the main results, we can draw the following observations:

**Effectiveness of *ReSearch*** Compared with all the baselines, *ReSearch* achieves significant improvements on all the benchmarks, which demonstrates the effectiveness of our proposed framework. Specifically, among all the benchmarks, the average improvement of *ReSearch* over the best baseline is **15.81%** in exact match and **17.56%** in LLM-as-a-judge, for Qwen2.5 model with 7B parameters. For Qwen2.5 model with 32B parameters, the average improvement is **14.82%** in exact match and **15.46%** in LLM-as-a-judge.

**Comparison between base and instruction-tuned models** We train *ReSearch* from both base and instruction-tuned models with 7B and 32B parameters respectively, and note that they are all trained using reinforcement learning from scratch without any supervised fine-tuning. From the results, we can observe training from the instruction-tuned model can further improve the performance of *ReSearch*. Such observation is consistent among all the benchmarks and model sizes.

**Generalization Ability** During reinforcement learning, *ReSearch* learns the ability of reasoning with search, which is independent of specific knowledge or multi-hop patterns, and such ability is generalizable. Our model *ReSearch* is only trained on the training set of MuSiQue dataset, but from the results, we can observe that it can generalize to other benchmarks with different question types and structures, which demonstrates the generalization ability of *ReSearch*.

## 3.3 Further Analysis

We investigate the important metrics during training *ReSearch* in this section. Specifically, the response length and number of search operations during training are shown in Figure 3 respectively.

Table 2: Exact Match (EM, %) and LLM-as-a-Judge (LJ, %) results on multi-hop question answering benchmarks. The best results are highlighted in bold, and the best results across baselines are underlined.

| Model | HotpotQA | | 2Wiki | | MuSiQue | | Bamboogle | |
|---|---|---|---|---|---|---|---|---|
| | EM | LJ | EM | LJ | EM | LJ | EM | LJ |
| **Qwen2.5-7B(-Instruct)** | | | | | | | | |
| Naive Generation | 19.18 | 30.64 | 25.76 | 27.87 | 3.76 | 10.38 | 10.40 | 22.40 |
| Naive RAG | 31.90 | 49.59 | 25.78 | 29.52 | 6.21 | 12.78 | 20.80 | 32.00 |
| Iter-RetGen | 34.36 | 52.22 | 27.92 | 31.86 | 8.69 | 16.14 | 21.60 | 35.20 |
| IRCoT | 30.33 | 52.06 | 21.57 | 30.65 | 6.99 | 14.19 | 24.80 | 36.80 |
| *ReSearch*-Qwen-7B | 40.57 | 60.26 | 44.67 | 50.06 | 21.68 | 32.19 | **43.20** | **54.40** |
| *ReSearch*-Qwen-7B-Instruct | **43.52** | **63.62** | **47.59** | **54.22** | **22.30** | **33.43** | 42.40 | **54.40** |
| **Qwen2.5-32B(-Instruct)** | | | | | | | | |
| Naive Generation | 24.63 | 38.26 | 27.23 | 29.68 | 6.12 | 14.23 | 18.40 | 29.60 |
| Naive RAG | 36.46 | 55.73 | 30.38 | 34.87 | 9.27 | 15.97 | 23.20 | 40.80 |
| Iter-RetGen | 39.81 | 58.80 | 33.64 | 38.22 | 12.49 | 20.11 | 29.60 | 44.80 |
| IRCoT | 28.44 | 55.44 | 13.53 | 29.50 | 7.82 | 18.20 | 31.20 | 47.20 |
| *ReSearch*-Qwen-32B | 42.77 | 64.27 | 38.52 | 45.59 | **26.40** | 37.57 | 54.40 | 66.40 |
| *ReSearch*-Qwen-32B-Instruct | **46.73** | **67.70** | **44.90** | **50.30** | **26.40** | **38.56** | **56.80** | **67.20** |

The curve of training reward and validation reward are shown in Figure 4. The validation is conducted on a part of development set of MuSiQue dataset with 100 random samples, and conducted every 10 steps during training.

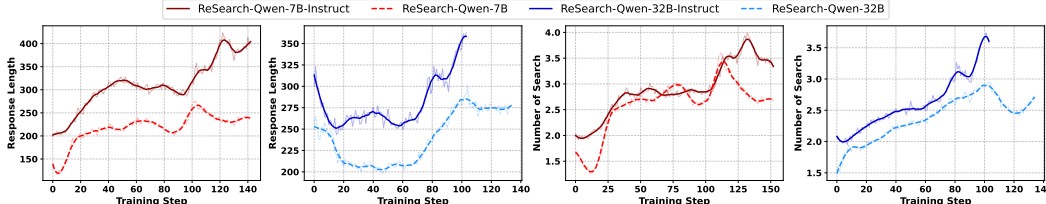

Figure 3: Response length and number of search operations during training.

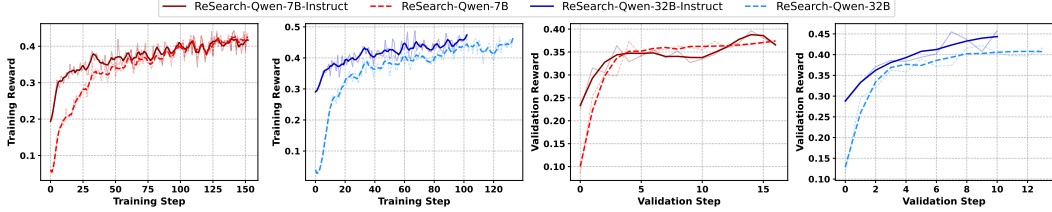

Figure 4: Training and validation reward during training.

**Response Length** We define the response length as the total number of tokens in a model's output, excluding retrieval results, which can be interpreted as the test-time cost of reasoning. From the first two figures in Figure 3, it is evident that the response length generally increases throughout the training process. Instruction-tuned models exhibit longer response lengths compared to base models for both the 7B and 32B parameters. Additionally, for the 32B models, the response length initially decreases during the first 20 training steps before increasing again after approximately the 60th step. This observed behavior may be attributed to the 32B models having more inherent knowledge, leading

them to generate longer responses initially, and do not utilize retrieval. However, after receiving some guidance from reward signals early in the training, they begin to leverage searches more frequently, thereby reducing their reliance on generated knowledge.

**Number of Search Operations**  We also count the average number of search operations in one rollout during training, which is shown in the last two figures in Figure 3. As depicted in the figures, the number of search operations shows consistent growth throughout the training process. Such trend indicates that for complex multi-hop questions, the model gradually learns to use search to retrieve relevant information iteratively multiple times.

**Training and Validation Reward**  We show the training and validation reward during reinforcement learning of *ReSearch* in Figure 4. The reward increases sharply during the first 20 training steps, and then gradually increases. The cold-start reward is higher for instruction-tuned models for both 7B and 32B models. For 7B models, the rewards converge to a similar level for both base and instruction-tuned models, while for 32B models, the training reward of instruction-tuned models is higher than that of base models.

## 3.4   Case Study

Table 3: A case study of *ReSearch* during training.

| |
|---|
| **Question:** Who won the 2018 presidential election in the country where the political party of Martín Ramírez Pech operates? 
 **Answer:** Andrés Manuel López Obrador |
| `<think>` To find the answer, I first need to determine the country in which Martín Ramírez Pech's political party operates. I will search for "Martín Ramírez Pech" to find out this information. `</think>` |
| `<search>` Martín Ramírez Pech `</search>` |
| `<result>` "Martín Ramírez", Martín Ramírez Martín Ramírez (January 30, 1895 – February 17, 1963) was ... ... \n "Martín Ramírez", and $134,500 in New York in 2011. Martín Ramírez Martín Ramírez (January 30, 1895 – ... ... \n "Martín Ramírez", including brown paper bags, scraps of examining-table paper, and book pages ... ... `</result>` |
| `<think>` It appears that I made a mistake in my search query. Instead of searching for "Martín Ramírez Pech", the results are showing information about a different "Martín Ramírez". To find out the right information, I need to look for Martín Ramírez Pech specifically and the political party associated with him. I will modify the search query to "Martín Ramírez Pech political party" for the correct information. `</think>` |
| `<search>` Martín Ramírez Pech political party `</search>` |
| `<result>` "Cuauhtémoc Blanco", of Morelos. In the 2015 legislative elections, he won in a closely contested election ... ... \n "Labor Party (Mexico)", for PT and 75 deputies and 16 senators for the PES. The alliance has received ... ... \n "Graco Ramírez", Graco Ramírez Graco Luis Ramírez Garrido Abreu (born 26 June 1949, in Villahermosa, Tabasco) is a Mexican ... ... `</result>` |
| `<think>` Now, we see that Martín Ramírez Pech is associated with the political party known as the Labor Party (PT) in Mexico. I now need to continue this line of reasoning to find out who won the presidential election in 2018 in Mexico. `</think>` |
| `<search>` 2018 Mexican presidential election `</search>` |
| `<result>` "2018 Mexican general election", (INE) officially declared the new process underway on 8 September 2017. The presidential election was won by Andrés Manuel López Obrador of ... ... \n "2018 Mexican general election", irregularities including reports of vote-buying, spending beyond legal campaign ... ... \n "2018 Mexican general election", 2018 Mexican general election General elections were held in Mexico on 1 July 2018 ... ... `</result>` |
| `<think>` According to the search results, the 2018 Mexican presidential election was won by Andrés Manuel López Obrador as the candidate for the National Regeneration Movement (MORENA). `</think>` |
| `<answer>` The final answer is \boxed{Andrés Manuel López Obrador} `</answer>` |

To give a more intuitive understanding of the utility of *ReSearch*, we show a case study in Table 3. This case is from the reinforcement learning process of Qwen2.5-32B-Instruct model. Where text enclosed by `<think>` and `</think>`, `<search>` and `</search>`, and `<answer>` and `</answer>` are generated by the model, and the text enclosed by `<result>` and `</result>` are retrieved from the retrieval environment. For clarity, we use "... ..." to represent the truncation of the retrieval results. From this case, we can see that the model can effectively break down the complex question and conduct reasoning within `<think>` and `</think>`. Such reasoning process is crucial for guiding when and what to search, and leading to the final answer in a multi-step manner.

**Self-elicited Reflection**   In addition, we also observe reflection phenomenon in the model's response. As depicted in the second thinking step in Table 3, the model states, "I made a mistake", recognizing that the previous search query failed to retrieve useful information. It then corrects itself in the third thinking step by generating a more effective search query to obtain the relevant information. Note that such reflection ability is not explicitly trained or designed in the prompt templates, but is naturally elicited from the model itself during reinforcement learning.

## 4   Related Work

### 4.1   Reinforcement Learning with LLMs

Reinforcement learning [30], which aims to maximize the expected return of an agent's policy through interactions with the environment, has emerged as a crucial technique for LLMs, from aligning with human values to enhancing reasoning capabilities. A significant development was Reinforcement Learning from Human Feedback (RLHF) [18], which uses Proximal Policy Optimization (PPO) [22] with reward models trained on human preferences. Several methods have since improved upon PPO, including Direct Preference Optimization (DPO) [20], Simulated Preference Optimization (SimPO) [15], and Group Relative Policy Optimization (GRPO) [24]. Recently, reinforcement learning has demonstrated remarkable success in enhancing reasoning capabilities, as evidenced by notable achievements such as OpenAI-o1 [17], DeepSeek-R1 [5], and Kimi k1.5 [31]. However, the reasoning capabilities of LLMs under RAG settings remain largely unexplored. Meanwhile, several concurrent works have also begun to investigate reinforcement learning for enhancing LLM reasoning with tool use [9, 29].

### 4.2   Retrieval-Augmented Generation for LLMs

RAG techniques [8, 12] augment LLMs by retrieving external knowledge and incorporating it into the generation process. Extensive research has been conducted in this area, encompassing various aspects such as retriever optimization [27, 38], query refinement [3], and self-reflection mechanisms [2]. For more complex scenarios, particularly in addressing multi-hop questions, iterative RAG models [23, 33] have been developed that alternately perform retrieval-enhanced generation and generation-enhanced retrieval. Additionally, supervised learning approaches have been explored using annotated trajectories of multi-step retrieval [21, 35]. Despite these advances, the application of reinforcement learning to enhance reasoning capabilities within RAG-style tool-augmented environments remains an open and under-explored research direction.

## 5   Conclusion

In this paper, we introduced *ReSearch*, a novel framework that trains LLMs to reason with search via reinforcement learning without requiring any supervised data on reasoning steps. Our approach integrates search operations as integral components of the reasoning chain, where text-based thinking guides when and how to perform searches, and search results subsequently influence further reasoning. Through extensive experiments on multiple multi-hop question answering benchmarks, we demonstrated that *ReSearch* achieves significant improvements over baseline methods. The results also indicate the framework's potential for more realistic scenarios. Analysis of the training process revealed that *ReSearch* naturally elicits advanced reasoning capabilities such as reflection and self-correction, without relying on pre-defined heuristics. This work highlights the effectiveness of integrating reasoning and search operations through reinforcement learning, offering a promising direction for developing more capable and reliable LLM-based systems for complex multi-hop tasks.

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

# A Prompt for LLM-as-a-Judge

```
Prompt for Extracting Scenarios

You will be given a question and its ground truth answer list
    where each item can be a ground truth answer. Provided a
    pred_answer, you need to judge if the pred_answer correctly
    answers the question based on the ground truth answer list.
    You should first give your rationale for the judgement, and
    then give your judgement result (i.e., correct or incorrect).

Here is the criteria for the judgement:
1. The pred_answer doesn't need to be exactly the same as any of
    the ground truth answers, but should be semantically same
    for the question.
2. Each item in the ground truth answer list can be viewed as a
    ground truth answer for the question, and the pred_answer
    should be semantically same to at least one of them.

question: {question}
ground truth answers: {gt_answer}
pred_answer: {pred_answer}

The output should in the following json format:
```json
{
    "rationale": "your rationale for the judgement, as a text",
    "judgement": "your judgement result, can only be 'correct'
        or 'incorrect'"
}
```

Your output:
```

# B Implementation Details

Our training is conducted on $8 \times 8$ Nvidia H800 GPUs, with full parameter optimization and gradient checkpointing. We show some important parameter settings in Table 4.

Table 4: Implementation details of *ReSearch*.

| Parameter | Value |
|---|---|
| Learning Rate | 1e-6 |
| Train Batch Size | 256 |
| Number of Training Epochs | 2 |
| Number of Rollout | 5 |
| Rollout Temperature | 1.0 |
| KL Loss Coefficient | 0.001 |
| Clip Ratio | 0.2 |

# C Limitation

While our work demonstrates promising results in training LLMs to reason with search, there are some limitations to consider. Our current framework primarily focuses on scenarios where the

answers are relatively concise and can be objectively verified through simple metrics like F1 score. This approach may not generalize well to tasks requiring longer, more nuanced responses, where more sophisticated reward modeling would be necessary to effectively guide the reinforcement learning process. Additionally, like many existing works in retrieval-augmented generation, our study utilizes Wikipedia as the primary knowledge base for RAG operations, following the common practice due to the availability of standardized open-source knowledge bases. This limitation means we have not yet explored the framework's effectiveness with other types of specialized or domain-specific knowledge bases that might be more appropriate for certain applications. Future work could investigate extending our approach to handle more complex response types and diverse knowledge sources beyond general encyclopedic knowledge.

## D  Broader Impact

Our work on *ReSearch* has several potential positive societal impacts. By improving the ability of LLMs to reason with search capabilities, this framework can enhance the accuracy and reliability of AI systems in knowledge-intensive tasks, particularly benefiting fields such as education, scientific research, and fact-checking. The framework's ability to break down complex questions into manageable steps while verifying information through external sources could help reduce the spread of misinformation and improve the quality of AI-assisted decision-making. Furthermore, the self-reflective capabilities that emerge during training could lead to more transparent and explainable AI systems, fostering greater trust between users and AI technologies. However, we acknowledge potential concerns that warrant consideration. The increased efficiency in information retrieval and processing could lead to higher computational resource consumption, potentially contributing to environmental impacts through increased energy usage. Additionally, while our framework improves accuracy, there remains a small possibility of reinforcing existing biases present in the search results or knowledge bases used for training. To mitigate these concerns, we recommend implementing energy-efficient training strategies and regularly auditing the search sources used in the system. Overall, we believe the benefits of more accurate, transparent, and reliable AI systems outweigh these manageable risks, particularly as the technology continues to evolve with appropriate safeguards and monitoring mechanisms in place.

