# OpenReview forum: "ReSearch: Learning to Reason with Search for LLMs via Reinforcement Learning"
_NeurIPS.cc/2025/Conference — NeurIPS 2025 poster_

### Official Review · Reviewer_KyAb · 2025-06-17

**Clarity:** 4
**Significance:** 3
**Originality:** 3
**Rating:** 5
**Confidence:** 4

**Summary:**

This work studies on LLM reasoning with search via reinforcement learning. To address the challenges, authors propose a novel framework, named ReSearch, which integrates the search operations into the reasoning process. Experiments on Qwen2.5-7B/32B models across several multiple-hop reasoning benchmarks (HotpotQA, 2Wiki, MusiQue and Bamboogle) demonstrate the effectiveness of the proposed method.

**Questions:**

How do you address the issue of growthly sequence length during multi-turn search in the training process? I understand that after several rounds of search, the concatenation of search results and generated reasoning tokens could become excessively long. Although the search results do not contribute to loss calculation, their cumulative expansion during iterative search still leads to substantial computational overhead. How is this challenge resolved in your approach?

**Ethical Concerns:**

["NO or VERY MINOR ethics concerns only"]

**Final Justification:**

My concerns are addressed. I will maintain my positive score.

**Limitations:**

No.

**Quality:**

3

**Strengths And Weaknesses:**

**Strengths**

1. The work studies on an interesting and important topic for agentic AI. Beyond naive "search everything" or "never search" approaches, the core question of when, how, and where to integrate search into the reasoning process is fundamental for building modern LLM agents. I think this work is the one of the earliest exploration in this field.

2. The proposed ReSearch method is well-motivated and novel. Building upon the naive GRPO, this work excels in the multi-turn interaction with search tools which is triggered by the special tokens, and retrieval result masking which enables policy model to focus only on the reasoning part. I appreciate the simple but effecitve design of the method.

3. Significant improvements across several multi-hop question answering tasks. Authors evaluates 7B(small) and 32B(large) models on four standard multi-hop QA benchmarks with insightful analysis on response length and number of search rounds. The experiments are comprehensive and convicing.

**Weaknesses**

1. Comparisons with prompt-based baselines and naive RAGs appear somewhat unfair. More robust baselines, such as DeepSeek-R1, should be considered for a more comprehensive evaluation.

2. The base models used are all non-reasoning models. Investigations into reasoning models would be highly valuable, including models like QwQ-32B-preview, the Qwen3-series, and DeepSeek-R1-Distilled-Qwen models.

---

> ### Author Rebuttal · Authors · 2025-07-31
>
> We sincerely appreciate the reviewer’s thoughtful comments and suggestions. Below, we address each of the weaknesses (W) and questions (Q) raised.
>
> ---
>
> **[W1] Comparisons with more robust baselines**
>
> While we did not directly run more robust baselines, such as DeepSeek-R1, we collected their evaluation results from other recent works [1] that follow the same benchmark datasets as ours. The exact match (EM) scores for DeepSeek-R1 and Qwen3-235B-A22B are taken from [1], while the results for ReSearch-Qwen2.5-7B-Instruct and ReSearch-Qwen2.5-32B-Instruct are from our own experiments. The comparison is shown below:
>
> | Model | HotpotQA | 2Wiki | MuSiQue | Bamboogle |
> |:-----------|:---------:|:-------:|:--------:|:---------:|
> | DeepSeek-R1|   48.0   | 54.0  |  27.5   |   52.0    |
> | Qwen3-235B-A22B |   44.5   | 45.3  |  27.6   |   43.8    |
> | ReSearch-Qwen2.5-7B-Instruct |   43.5   | 47.6  |  22.3   |   42.4    |
> | ReSearch-Qwen2.5-32B-Instruct|   46.7   | 44.9  |  26.4   |   56.8    |
>
> It is evident that, although our models are considerably smaller than DeepSeek-R1 and Qwen3-235B-A22B, their performance is overall comparable across these challenging multi-hop benchmarks.
>
> ---
>
> **[W2] Investigation of training on reasoning bases**
>
> Thank you for highlighting the potential value of evaluating ReSearch on models that already contain some form of chain‑of‑thought reasoning. Our current study deliberately starts from non‑reasoning checkpoints for two reasons:
>
> 1. Clean attribution.  Starting from a “plain” model lets us see exactly how much reasoning‑with‑search our RL pipeline adds, without mixing in skills learned elsewhere.
>
> 2. Compute and openness.  Reasoning‑enhanced models (e.g., DeepSeek‑R1‑Distilled‑Qwen, >30 B params) are heavyweight, use private data, and are costly to reproduce. In contrast, vanilla Qwen‑2.5 is fully open and fits within our compute budget.
>
> This keeps the study controlled and reproducible. We agree that plugging ReSearch into QwQ‑32B‑preview, Qwen3, or distilled DeepSeek variants is a promising next step. We will release code hooks so that the community can replicate these experiments.
>
> ---
>
> **[Q1] Issue of growing sequence length**
>
> We did not introduce any truncation or external‑memory mechanism in the current study. As described in §3.1, the retriever returns at most the top‑5 passages per query, keeping each search block to only a few hundred tokens; in practice, the full context seldom approaches the model’s 32 k‑token window. Figure 3 further shows that the reasoning tokens remain well below this limit across multiple turns, so the computation comfortably fits within our hardware budget.
>
> We acknowledge, however, that context growth becomes a real bottleneck for longer‑horizon tasks. Recent work suggests two complementary remedies: (i) treating the thinking content as a working memory to retain important reasoning or facts on previous steps [2]; and (ii) summarizing each retrieved passage before appending it, preventing unbounded expansion [3]. These techniques are orthogonal to ReSearch, and we plan to integrate them in future work.
>
> ---
>
> *References*
>
> [1] DynaSearcher: Dynamic Knowledge Graph Augmented Search Agent via Multi-Reward Reinforcement Learning, 2025.
>
> [2] MEM1: Learning to Synergize Memory and Reasoning for Efficient Long‑Horizon Agents, 2025.
>
> [3] WebSailor: Navigating Super‑Human Reasoning for Web Agents, 2025.

---

> ### Comment · Reviewer_KyAb · 2025-08-04
>
> Thanks for authors' reponse. My concerns are addressed. I will maintain my positive score.

---

> > ### Author Response · Authors · 2025-08-04
> >
> > Thank you for your considerate response and support. We appreciate your constructive comments and will take them into account to enhance our work in the final version.

---

### Official Review · Reviewer_75QP · 2025-06-23

**Clarity:** 3
**Significance:** 3
**Originality:** 3
**Rating:** 6
**Confidence:** 4

**Summary:**

This paper presents ReSearch, a reinforcement learning framework that enables LLMs to reason with external search in a unified, text-driven manner. Unlike prior approaches that rely on supervised reasoning steps, ReSearch treats search as an integral part of the reasoning process—learning when and how to search through natural language thought.

The method is trained on Qwen2.5 models using only one dataset, yet demonstrates strong generalization across multiple multi-hop QA benchmarks. The authors also observe emergent behaviors such as reflection and self-correction during training.

**Questions:**

1. As mentioned above, I'm particularly curious about whether this pure RL training approach can achieve stable improvements on more challenging benchmarks such as BrowseComp and BrowseComp-ZH, as well as the potential challenges it might encounter.

2. How do the performance comparisons look across different training strategies: pure RL, pure SFT, and SFT+RL combinations?

3. Have you explored any scaling laws for this task?

**Ethical Concerns:**

["NO or VERY MINOR ethics concerns only"]

**Final Justification:**

The authors provide a reasonable justification for focusing on Wikipedia-only retrieval to isolate the effects of RL. Their discussion of BrowseComp highlights meaningful challenges and outlines clear future directions. The comparison with STaR-style SFT is appropriate and supports their claims about RL benefits. Overall, the response adequately addresses the main concerns.

**Limitations:**

yes

**Paper Formatting Concerns:**

No formatting issues.

**Quality:**

4

**Strengths And Weaknesses:**

Strengths:
1. This paper trains ReSearch on different scales of the Qwen2.5 models, using only the MusiQue dataset for training. Despite this, it achieves performance gains across multiple benchmarks, demonstrating strong generalization capabilities.
2. The training framework is relatively straightforward and does not rely on supervised data. And the codebase is easy to understand and follow.

Weaknesses:
1. The evaluation lacks coverage of more complex benchmarks, such as BrowseComp and BrowseComp-ZH, which limits the assessment of the model's performance in more challenging scenarios.

---

> ### Author Rebuttal · Authors · 2025-07-31
>
> We sincerely appreciate the reviewer's efforts and insightful comments to improve our manuscript. Below, we address the weaknesses (W) and questions (Q) raised.
>
> ---
>
> **[Q1 & W1] On more challenging benchmarks**
>
> We share the reviewer’s interest in the performance of ReSearch on full‑web agent benchmarks such as BrowseComp.
>
> Our current study confines itself to Wikipedia retrieval so that we can cleanly measure how RL decides when to search versus when to keep thinking without additional confounders. BrowseComp, in contrast, exposes a much richer environment: the agent must issue DOM clicks, scroll, fill forms, and render JavaScript. Although a few recent systems have reported BrowseComp results by leveraging high-level “search + summary” APIs, this action space is still well beyond the "think, search" interface used here.
>
> The good news is that the key skill ReSearch learns—deciding when to query and when to reason internally—is orthogonal to the concrete low-level actions. In principle, the learned policy can sit atop any web-navigation layer once one is available. Nevertheless, several obstacles must be overcome before we can expect stable gains:
> - Richer action repertoire.  BrowseComp introduces clicks, back/forward, scrolling, and form submission. A naive flat policy would face an intractably large exploration space; hierarchical or curriculum RL is required.
> - Much longer horizons with sparse rewards.  Solving a BrowseComp question often involves ten or more navigation steps and yields only a single binary reward. Without intermediate shaping signals, the advantage estimates become extremely noisy.
> - Context bloat.  Raw HTML plus extracted text quickly exceed the model’s capacity unless working-memory or on-the-fly summarization is introduced.
>
> We have begun addressing these issues and outline three concrete extensions for future work:
> - Action layer.  We are integrating a lightweight DOM-navigation wrapper so the agent can click and scroll rather than issuing only textual queries.
> - Memory and compression.  We plan to add working-memory and retrieval-result summarization to control context length during long-horizon interactions.
> - Complex task generation.  We will create harder, multi-step question sets comparable to BrowseComp to stress-test the upgraded agent.
>
> We'll address these issues in our future work.
>
> ---
>
> **[Q2] Performance across different traning startegies**
>
> For an SFT baseline, we reproduced a STaR-style [1] pipeline as follows: (i) use Qwen-2.5-7B-Instruct to sample five rollouts for every MuSiQue training query (same prompt as our main experiments); (ii) keep only the trajectories whose final answer matches the ground truth; and (iii) fine-tune Qwen-2.5-7B on the resulting nearly 15k trajectories. Results show that this filtered-SFT model improves over prompt-only baselines but is consistently outperformed by our RL-trained ReSearch agent, underscoring the added value of RL.
>
> | Method     | HotpotQA | 2Wiki | MuSiQue | Bamboogle |
> |:---------|:----------:|:------:|:-------:|:---------:|
> | Qwen2.5-7B (STaR-style SFT) | 33.83 | 33.76 | 14.81 | 35.80 |
> | Qwen2.5-7B (ReSearch) | 40.57 | 44.67 | 21.68 | 43.20 |
>
> ---
>
> **[Q3] Scaling Laws for this task**
>
> Since our end‑to‑end RL + retrieval training is compute‑intensive, we restricted experiments to two backbone sizes (7 B, 32 B) and a fixed rollout budget, so we did not pursue formal scaling‑law analysis. We agree that understanding how performance scales with data, compute, and model size is crucial for tool‑augmented agents and plan to investigate these factors in future work; we will note this limitation in the revision.
>
> ---
>
> *References*
>
> [1] STaR: Bootstrapping Reasoning With Reasoning, 2022.

---

> > ### Comment · Reviewer_75QP · 2025-08-05
> > **Raising score**
> >
> > The authors provide a reasonable justification for focusing on Wikipedia-only retrieval to isolate the effects of RL. Their discussion of BrowseComp highlights meaningful challenges and outlines clear future directions. The comparison with STaR-style SFT is appropriate and supports their claims about RL benefits. While a formal scaling law study is missing, the limitations are acknowledged. Overall, the response adequately addresses the main concerns. I have raised my score.

---

> > > ### Author Response · Authors · 2025-08-06
> > >
> > > Thank you for your thoughtful review and positive assessment. We value your suggestions and will address your feedback in the next version of our paper.

---

### Official Review · Reviewer_XkNw · 2025-07-03

**Clarity:** 2
**Significance:** 3
**Originality:** 3
**Rating:** 4
**Confidence:** 3

**Summary:**

This paper introduces a framework to train LLMs to reason with search tools. To achieve this, the proposed framework applies reinforcement learning without process supervision. Experiment results show the effectiveness of the method with strong generalizability.

**Questions:**

see above.

**Ethical Concerns:**

["NO or VERY MINOR ethics concerns only"]

**Final Justification:**

This is a timely work that provides many important details for the community and the author also supplement insightful comments in the discussion period which helps progress the field.

**Limitations:**

Although the paper train model to answer open-ended question, the answers of these question are relatively short and subjective, making them easy to be evaluate. More complicated scenarios should cover longer answers and more complicated evaluation rubrics.

**Paper Formatting Concerns:**

not found

**Quality:**

2

**Strengths And Weaknesses:**

Strengths:
1. This paper study a timely topic of training reasoning models to use tools and answer open-ended question. The provided technical details are helpful for the community to learn how to build LLM applications.
2. The results demonstrate the effectiveness of the method. In particular, extensive evaluation of various benchmarks for single training show promising generalizability.
3. The introduced method is very simple, making it attractive for future exploration.

Weaknesses:
1. This paper lacks analysis on how different technical details affect the effect of the method. Given that the data used to train and evaluate in this study contain answers that are easy to evaluate, it is unclear potential challenges for researchers to study more complicated scenarios such as writing tech report.
2. Although the author make comparison to baseline methods such as native RAG, it is unclear the specific cases where the reasoning models equipped show advantages. It is unclear how training such reasoning agents bring improvement in these search tasks in specific. And what might previous method failed to cover.
3. The effeciency of the proposed framework in training and deployment in unclear.

---

> ### Author Rebuttal · Authors · 2025-07-31
>
> We sincerely appreciate the reviewer's efforts and insightful comments to improve our manuscript. Below, we address the weaknesses (W) raised.
>
> ---
>
> **[W1.1] How different technical details affect the effect of the method**
>
> Our method is intentionally minimalist, and all other components follow standard settings. Consequently, there are relatively few knobs to ablate. Nevertheless, we did run the most informative comparison we could devise—pure SFT vs. our RL training—to isolate the benefit of exploration. We give a STaR-style [1] SFT baseline:
>
> (i) Use Qwen-2.5-7B-Instruct to sample five rollouts for every MuSiQue training query (same prompt as our main experiments); (ii) keep only the trajectories whose final answer matches the ground truth; and (iii) fine-tune Qwen-2.5-7B on the resulting nearly 15k trajectories. Results show that this filtered-SFT model improves over prompt-only baselines but is consistently outperformed by our RL-trained ReSearch agent, underscoring the added value of RL.
>
> | Method     | HotpotQA | 2Wiki | MuSiQue | Bamboogle |
> |------------|----------:|------:|--------:|----------:|
> | Qwen2.5-7B (STaR-style SFT) | 33.83 | 33.76 | 14.81 | 35.80 |
> | Qwen2.5-7B (ReSearch) | 40.57 | 44.67 | 21.68 | 43.20 |
>
> The resulting filtered-SFT model does outperform prompt-only baselines, but it is consistently weaker than our RL-trained ReSearch agent on all four benchmarks. This experiment confirms that the extra exploration and credit assignment provided by RL—not merely the presence of chain-of-thought data—drives the performance gains.
>
> **[W1.2] Study more complicated scenarios**
>
> We deliberately began with tasks whose answers admit a rule‑based F1 reward so we could cleanly test whether reinforcement‑trained reasoning + search works at all, without confounding factors from noisy evaluators. We agree that extending to open‑ended outputs (e.g., multi‑page tech reports) demands richer feedback signals. Our framework is agnostic to the reward source, and in ongoing work we are replacing the string‑match reward with (i) rubric‑guided LLM judges and (ii) learned reward models that score content quality and coherence. We will add this discussion to the revision.
>
> ---
>
> **[W2] Where the reasoning agent shows advantages**
>
> *(1) Decomposition & planning for complex questions.*
>
> Many queries require breaking a goal into ordered sub‑goals and scheduling searches accordingly—something native RAG/prompt baselines do not do. Example (abridged):
>
>  “Name the famous bridge in the city where the composer of La costanza trionfante degl’amori e de gl’odii was born.”
> ReSearch explicitly: (i) finds the composer (Antonio Vivaldi), (ii) locates his birthplace (Venice), and (iii) retrieves the bridge (Rialto Bridge). A naïve RAG baseline, lacking this plan, often latches onto an unrelated bridge (e.g., Bridge of Sighs).
>
> *(2) Iterative correction when the first retrieval is wrong.*
>
> As shown in our case studies (Sec. 3.4), when an early search is off-target, ReSearch can detect the mismatch, revise its plan, and issue alternative queries. Baselines typically accept the first hit, propagating early errors with no recovery mechanism.
> These two capabilities—explicit multi-step planning and self-correction—explain why training a reasoning agent improves search tasks: it closes the loop between “think → search → verify → adjust,” which prior methods do not cover. We will make these points clearer in the revision.
>
> ---
>
> **[W3] Training and Deployment Efficiency**
>
> As specified in Appendix B, we clarify that our training experiments were conducted on 8 nodes × 8 Nvidia H800 GPUs (64 H800s in total). For deployment/inference, we use a single node with 4 H800 GPUs. We will make this information more explicit in the paper and further emphasize the practical deployment efficiency of our framework.
>
> ---
>
> *References*
>
> [1] STaR: Bootstrapping Reasoning With Reasoning, 2022.

---

> ### Comment · Reviewer_XkNw · 2025-08-05
>
> Thank you for you responses. Given this is a paper relating to timely topic, I appreciate the technical details and observations provided.
>
> It is surprising to see obvious gaps between SFT and RL results. I am curious about the computation resources required between the two settings. And whether SFT improves by scaling more data or data filtering. And how about doing iterative SFT? The author only study STaR like SFT while it would be more reasonable to compare RL to rejective finetuning.
>
> The author attributes this to extra exploration and credit assignment provided by RL which seems ambiguous. Why RL provides extra exploration than your rejective finetuning baseline? How GRPO provides credit assignment without a value model?

---

> > ### Author Response · Authors · 2025-08-06
> >
> > We sincerely thank the reviewer for their thoughtful follow-up and constructive feedback. Below, we address the specific concerns (C) you raised in your response.
> >
> > ---
> >
> > **[C1] Computation resources required between the two settings**
> >
> > *SFT baseline:*
> >
> > - We first perform reject sampling to keep only answer-correct trajectories – this runs on a single 8-H800 node for ~2 hours.
> > - Fine-tuning the 7B model on the filtered data with 2 epochs then takes another 1–2 hours on the same 8 GPUs.
> > - End-to-end, the entire SFT pipeline therefore costs ~3–4 wall-clock hours or 24–32 GPU-hours.
> >
> > *Our ReSearch (GRPO RL run):*
> >
> > - We train on-policy with 5 rollouts per query, using 8 nodes × 8 H800 (64 GPUs) in parallel.
> > - The run converges after ~5 wall-clock hours, totalling ~320 GPU-hours.
> >
> > Hence RL uses roughly 10 × more raw GPU-hours than the SFT pipeline. The longer wall-clock time for RL mainly reflects greater pipeline complexity compared to SFT, such as repeated rollouts and parameter transfers between inference and training. However, as frameworks in both industry and academia improve—especially with asynchronous rollouts and better engineering, the efficiency gap is rapidly shrinking. We expect RL training to become much faster over time, without losing its core advantages.
> >
> > ---
> >
> > **[C2] Whether SFT improves by scaling more data or iterative rejective finetuning**
> >
> > We conducted further experiments to assess whether SFT performance can be improved by iterative rejective fine-tuning or simply scaling up the filtered data. Specifically, we ran three rounds of STaR-style SFT (with filtered data sizes of 15,675; 23,025; and 26,571), and also an "ALL" experiment using all rounds' data combined. In each round of our iterative SFT experiments, we used the SFT model from the previous iteration to perform reject sampling on the original MuSiQue training set—the same data source used for RL training—to ensure a fair comparison.
> > As shown below, both iterative and scaled SFT bring only limited gains, and the improvements quickly plateau. By contrast, our RL-based ReSearch model consistently outperforms all SFT variants.
> >
> > | Method     | HotpotQA | 2Wiki | MuSiQue | Bamboogle |
> > |------------|:--------:|:------:|:------:|:-------:|
> > | Qwen2.5-7B (STaR-style SFT Iter1) | 33.83 | 33.76 | 14.81 | 35.80 |
> > | Qwen2.5-7B (STaR-style SFT Iter2) | 33.50 | 37.06 | 16.05 | 37.60 |
> > | Qwen2.5-7B (STaR-style SFT Iter3) | 33.39 | 34.76 | 16.30 | 36.00 |
> > | Qwen2.5-7B (STaR-style SFT All) | 33.65 | 36.68 | 16.09 | 40.00 |
> > | Qwen2.5-7B (ReSearch) | 40.57 | 44.67 | 21.68 | 43.20 |
> >
> > ---
> >
> > **[C3] RL vs. SFT**
> >
> > SFT trains on a static set of already-successful trajectories, so the model can only imitate behaviour that is known to work. RL, in contrast, continually samples fresh rollouts and updates on both high-reward and low-reward trajectories, teaching the policy to distinguish helpful from harmful actions. As a result, SFT tends to memorise familiar reasoning patterns, whereas RL generalises better to unseen problems, a difference also highlighted in prior work [1].
> >
> > For credit assignment, our earlier description may not have been sufficiently clear. In our work, credit assignment specifically refers to sequence-level (trajectory-level) assignment based on the outcome reward, rather than per-token credit assignment as in PPO with a value model. In GRPO, each trajectory receives a scalar reward, and all tokens in the trajectory are updated according to the overall outcome—positively for good rollouts and negatively for bad ones. This enables the model to learn from both successful and unsuccessful attempts, unlike SFT, which only imitates positive examples.
> >
> > ---
> >
> > *References:*
> >
> > [1] SFT Memorizes, RL Generalizes: A Comparative Study of Foundation Model Post-training

---

> > ### Author Response · Authors · 2025-08-09
> >
> > Thank you again for your constructive feedback on our work. We have responded to your concern in detail and would greatly value any further thoughts you might have. As the discussion period will close on Aug 8, 11:59 pm AoE, we hope you might have a chance to review our reply before then.

---

### Official Review · Reviewer_NC8m · 2025-07-04

**Clarity:** 3
**Significance:** 2
**Originality:** 2
**Rating:** 4
**Confidence:** 4

**Summary:**

This paper proposes using GRPO training for multi-hop QA  datasets, which require search for answering. Since outcome-level reward is utilized for RL training (based on answer accuracy and partial credit for format adherence), the proposed ReSearch framework does not require step-level labelled reasoning data, which is very expensive, and requires substantial human effort for generation. This method considers search as an integral component of generating the reasoning for multi-hop QA tasks and aims to guide the search procedure (when and where to search) by using text-level thinking steps. Empirical results show the effectiveness of Research compared to standard RAG and some existing test-time prompt based approaches.

**Questions:**

Please address the issues raised in the 'Weaknesses' section above. I will be happy to revise my score if those concerns are sufficiently alleviated during the rebuttal. Score updated after the rebuttal.

**Ethical Concerns:**

["NO or VERY MINOR ethics concerns only"]

**Final Justification:**

I have carefully read the other reviews and the authors' rebuttals. Overall, my concerns are addressed adequately, so I am raising my score to reflect that.

**Limitations:**

Discussed adequately in Appendices C and D.

**Paper Formatting Concerns:**

There are some minor typos, e.g. 'lableled' in line 74, 'conduct' (--> conducted) in line 413 etc.

**Quality:**

3

**Strengths And Weaknesses:**

Strengths:

1) ReSearch aims for the LLMs to learn when and where to search and how to use the result to generate the next thinking step via RL. This is an interesting avenue for tackling complex, multi-hop QA problems and has some novelty compared to standard prompting based and/or agentic approaches. RL training also encourages self-reflection and self-correction. Using GRPO eliminates the need of having a critic network (although that aspect is not the contribution of this work and is inherent to GRPO algorithm).

2) ReSearch is trained on a single multi-hop dataset yet demonstrates strong performance on diverse, standard benchmarks; HotpotQA, 2Wiki, MuSiQue, and Bamboogle.

3) The paper is clear and easy to read. Sufficient details are provided for a reader to understand the technique proposed.

Weaknesses:

1)  All the baselines considered in this work are prompt based. There is no comparison to any other method which require either RL training or SFT. In particular, if the comparison with an SFT approach is not presented, how can we understand the usefulness of the RL pipeline used here? One could use the same rollouts used in this paper to do an SFT after filtering the data based on the correctness of the final answer (as in the STAR algorithm, https://arxiv.org/pdf/2203.14465). Can GRPO outperform such a simple SFT strategy?

2)  Despite integration of search steps, rewards are still based on final answer correctness and instruction following capability (outcome-level), not the quality of the search decisions. This may lead to suboptimal or inefficient search patterns (unjustified calls or missing retrievals) because there is no intermediate reward signal assessing search utility. Given that process-level reward modelling is better than outcome-level rewards in other domains (e.g. math and logical reasoning), is there a way to extend ReSearch to utilize process rewards?

3) Some discussion (and comparison to) of recent related work in the same area (e.g. R-Search (https://www.arxiv.org/pdf/2506.04185) and Search-R1 (https://arxiv.org/pdf/2503.09516)) is required for delineating the novelty of this work. Both of these aforementioned works tackle the exact same task (multi-hop QA with search) and use RL training. A careful methodological comparison between ReSearch and these papers would strengthen this paper.

4) There is no MCTS style baseline considered. Given the nature of these tasks, test-time compute using tree-structured heuristic search could be another potential avenue to improve performance. Are there any existing baseline which does that in this domain? If there is any, the RL training approaches need to be compared against such test-time methods.

---

> ### Author Rebuttal · Authors · 2025-07-31
>
> We sincerely appreciate the reviewer's efforts and insightful comments to improve our manuscript. Below, we address the weaknesses (W) raised.
>
> ---
>
> **[W1] Comparison with simple SFT strategies**
>
> *Why RL beats plain SFT*
>
> Supervised fine-tuning (SFT) can only imitate trajectories that the current model already happens to generate; it cannot move beyond the quality of those teacher samples. Reinforcement learning (RL) instead allows on-policy exploration: the model tries new tool-calling sequences, receives scalar rewards, and updates itself accordingly. This feedback loop teaches the model not only what a good trajectory looks like but also why it is better than the alternatives—crucial for tool-augmented agents, where many useful reasoning paths are so unlikely under an untrained policy that they never appear in an SFT dataset.
>
> *Our STaR-style SFT baseline*
>
> To give SFT its best shot, we reproduced a STaR-like pipeline: (i) use Qwen-2.5-7B-Instruct to sample five roll-outs for every MuSiQue training query (same prompt as our main experiments); (ii) keep only the trajectories whose final answer matches the ground truth; and (iii) fine-tune Qwen-2.5-7B on the resulting nearly 15k trajectories. Results show that this filtered-SFT model improves over prompt-only baselines but is consistently outperformed by our RL-trained ReSearch agent, underscoring the added value of RL.
>
> | Method     | HotpotQA | 2Wiki | MuSiQue | Bamboogle |
> |:---------|:----------:|:------:|:-------:|:---------:|
> | Qwen2.5-7B (STaR-style SFT) | 33.83 | 33.76 | 14.81 | 35.80 |
> | Qwen2.5-7B (ReSearch) | 40.57 | 44.67 | 21.68 | 43.20 |
>
> We intentionally did not reuse rollouts generated during GRPO training, because those samples are themselves produced by an ever-improving policy—a that benefits inherent to RL.  Using a static model (here we use Qwen2.5-7B-Instruct) keeps the comparison faithful to standard SFT practice.
>
> ---
>
> **[W2] Extending ReSearch with process‑level rewards**
>
> While our current design deliberately uses a simple, cheap reward (answer correctness + format) to avoid heavy annotation, we agree that ReSearch can be naturally extended to incorporate process signals. Concretely, the following heuristics can be injected into our framework without altering the GRPO loop:
>
> - Redundancy penalty: If a retrieval step returns passages that highly overlap with previously retrieved content, assign a negative reward to discourage wasted calls and encourage informational diversity.
>
> - Self-critique feedback: Use a lightweight verifier to flag unsupported or logically inconsistent intermediate statements, yielding sparse negative rewards.
>
> These additions keep our RL loop intact (still GRPO) and only modify the reward function. We will add a short discussion of the above options and leave them as future work, acknowledging that richer process signals are orthogonal and complementary to our current outcome-level scheme.
>
> ---
>
> **[W3] Discussion of R‑Search & Search‑R1 (contemporaneous work)**
>
> We appreciate the pointer. Both R‑Search (June 2025) and Search‑R1 (March 2025) were released after March 1, 2025 and thus fall under NeurIPS’s “contemporaneous work” policy. Per the call for papers, such work should be cited and discussed but the absence of full empirical comparison is not grounds for rejection.
>
> Methodologically, ReSearch differs along several axes:
>
> - Reward design: We use purely outcome-level GRPO—without step labels, PRMs, or auxiliary verifiers. In contrast, R‑Search incorporates process-level rewards or external evaluators (such as cross-family reward models for evidence quality). Furthermore, Search‑R1 adopts rule-based rewards based on exact string matching, whereas our work uses F1‑score as a smoother and more informative reward signal compared to exact match.
> - Model scale. We report results on larger backbones (7B and 32B), whereas R‑Search/Search‑R1 primarily train/evaluate 3B–7B models.
>
> We will add a concise paragraph to clearly position ReSearch relative to these contemporaneous papers.
>
> ---
>
> **[W4] Comparison to MCTS-style test-time search**
>
> ReasonRAG [1] integrates Monte Carlo Tree Search (MCTS) into an agentic RAG pipeline to offline-generate process‑supervised data (step scores) before training. It is thus a close, tree‑search–based counterpart in this domain. Under the same evaluation protocol (Qwen2.5‑7B‑Instruct backbone, Exact Match metric), ReSearch consistently outperforms ReasonRAG on all four benchmarks:
>
> | Method     | HotpotQA | 2Wiki | MuSiQue | Bamboogle |
> |:------------|:---------:|:------:|:------:|:--------:|
> | ReSearch   | 43.5     | 47.6 | 22.3   | 42.4     |
> | ReasonRAG  | 38.4     | 43.6 | 12.8   | 36.0     |
>
> These results suggest that reinforcement learning with a simple outcome reward can match—and even surpass—the performance of more complex MCTS-based supervision.
>
> ---
>
> *References*
>
> [1] Process vs. Outcome Reward: Which is Better for Agentic RAG Reinforcement Learning. 2025

---

> > ### Comment · Reviewer_NC8m · 2025-08-04
> > **raising score**
> >
> > I thank the authors for their thorough rebuttal, which addressed my concerns. I have raised my score to 4.

---

> > > ### Author Response · Authors · 2025-08-04
> > >
> > > Thank you very much for your thoughtful review and positive evaluation. We will carefully reflect your suggestions and feedback in the next version of our paper. We appreciate your support and valuable input.

---

### Comment · Area_Chair_oFKE · 2025-08-05
**Engagement with Rebuttal**

All,

Thanks to every reviewer for logging in and engaging with the review. It appears that every author has discussed this paper so far and I'd like to encourage everyone to engage and wrap up by the new deadline of Aug 8, 11.59pm AoE.

---

### Decision · Program_Chairs · 2025-09-17

**Decision:**

Accept (poster)

**Comment:**

After a review round and a rebuttal where the authors and all reviewers engaged with the reviews there is a unanimous decision that this paper is technically solid and makes a good contribution to the research literature. Therefore this paper should be accepted to NeurIPS.

Key strengths include:

* Good articulation of the need for search when answering queries for multi-hop QA.

* Better performance than several benchmarks by levering good querying strategies, resulting in fewer queries and better performance.

Note that all reviews contain questions and guidance for updating and refining the paper and we strongly encourage the authors to take these into account with revising the paper.